# Measuring Eco-Anxiety with the Polish Version of the 13-Item Hogg Eco-Anxiety Scale (HEAS-13): Latent Structure, Correlates, and Psychometric Performance

**DOI:** 10.3390/healthcare12222255

**Published:** 2024-11-12

**Authors:** Paweł Larionow, Julia Mackiewicz, Karolina Mudło-Głagolska, Maciej Michalak, Monika Mazur, Magdalena Gawrych, Kamilla Komorowska, David A. Preece

**Affiliations:** 1Faculty of Psychology, Kazimierz Wielki University, 85-064 Bydgoszcz, Polandmudlo@ukw.edu.pl (K.M.-G.);; 2Independent Researcher, 2500 Valby, Denmark; 3Institute of Psychology, The Maria Grzegorzewska University, 02-353 Warsaw, Poland; 4Faculty of Health Sciences, School of Population Health, Curtin University, Perth, WA 6102, Australia; 5School of Psychological Science, The University of Western Australia, Perth, WA 6009, Australia; 6Department of Psychology, The School of Humanities and Sciences, Stanford University, Stanford, CA 94305, USA

**Keywords:** anxiety, climate anxiety, climate change worry, depression, eco-anxiety, environmental concerns, pro-environmental behaviours, psychometric properties, questionnaire, well-being

## Abstract

Background/Objectives: The Hogg Eco-Anxiety Scale (HEAS-13) is a thirteen-item measure of eco-anxiety, with four dimensions: (1) affective symptoms, (2) rumination, (3) behavioural symptoms, and (4) anxiety about personal impact. Being a recently developed questionnaire, data on its psychometrics are limited. The aim of this study was to introduce a Polish version of the HEAS-13 and examine its psychometric properties. Methods: Our sample consisted of 634 Polish-speaking adults, with ages ranging from 18 to 67 years. We assessed the HEAS-13’s factor structure, internal consistency, test–retest reliabilities, and its concurrent validity via relationships with climate-related variables, psychopathology symptoms, and well-being. We put emphasis on examining the discriminant validity of the HEAS-13 against general psychological distress. Results: As expected, the Polish HEAS-13 demonstrated strong factorial validity with an intended four-factor structure. The internal consistency and test–retest reliabilities of the scale were good and moderate, respectively. Higher levels of eco-anxiety were associated with higher environmental concerns, the experience of climate change (i.e., one’s perception of being affected by climate change), pro-environmental behavioural engagement, climate change worry, anxiety, and depressive symptoms, as well as lower levels of well-being. We empirically supported the strong discriminant validity of the HEAS-13, demonstrating that eco-anxiety was separable from general psychological distress. We also noted that females, younger people, and those with lower educational levels experienced higher eco-anxiety. To facilitate the use of this measure, we propose a potential screening cut-off value for the HEAS-13, which can indicate meaningfully elevated levels of eco-anxiety. Conclusions: Overall, the Polish version of the HEAS-13 has strong psychometric properties, usefully enabling the examination of climate-related anxiety. Our findings highlight its potential in cross-cultural research and healthcare practice.

## 1. Introduction

In recent decades, humanity has started facing climate change, considered one of the most urgent challenges for the planet [1]. Rising temperatures, changes in precipitation, extreme weather events, and other impacts of climate change have significant consequences for the environment, human health, and the global economy. At the same time, growing concerns and anxiety related to these issues are becoming more and more common across societies, affecting people’s functioning and well-being [2,3,4,5] as well as their pro-environmental behaviours [6].

### 1.1. The Eco-Anxiety Construct and Its Correlates

The concept of eco-anxiety encompasses “climate change anxiety” (i.e., fear related explicitly to anthropogenic climate change, including global warming, rising sea levels, and the increased incidence of natural disasters and extreme weather events) [7] and concerns about numerous environmental disasters that may or may not be directly caused by climate change (e.g., the elimination of entire ecosystems, plant and animal species, global mass pollution, and deforestation) [8]. A growing body of literature demonstrates the dual nature of eco-anxiety, which manifests, on the one hand, in harmful effects on mental health [9] and, on the other hand, in positive pro-environmental actions [10,11]. Metaphorically, it can be seen as a “double-edged sword”. As a result of its complex impact, discussions around eco-anxiety are frequently featured in modern media, and the phenomenon itself is gaining increased research interest [12].

Eco-anxiety is related to a wide range of psychopathologies [13], and its symptoms can vary in intensity from mild to more intense along a spectrum [14]. Although eco-anxiety is currently not considered a clinical disorder [15], a mounting body of evidence has shown that eco-anxiety is associated with higher mental ill-being (e.g., depression symptoms) and lower psychological well-being [7,9,15,16,17,18,19]. Previous studies have indicated that eco-anxiety can mediate the link between emotional regulation difficulties and worry about the future [20] or that higher climate anxiety is related to higher loneliness [21]. Such findings highlight that eco-anxiety may be a significant correlate for general negative mental health conditions.

Eco-anxiety is also associated with a variety of other climate-related constructs, such as different environmental concerns (i.e., egoistic, altruistic, and biospheric concerns) [22], the experience of climate change (i.e., a people’s level of recognition of being affected by climate change) [7,23], pro-environmental behaviour (e.g., recycling, turning off lights), and climate change worry (i.e., ruminations about climate change) [24].

### 1.2. The Hogg Eco-Anxiety Scale

In response to the growing interest in eco-anxiety, there is a need for measures that allow for a comprehensive assessment of this phenomenon and its impact on people. To study this, Hogg et al. [8] employed a mixed-methods approach to develop a comprehensive psychometric measure of eco-anxiety, resulting in a 13-item self-report questionnaire, the Hogg Eco-Anxiety Scale (HEAS-13) [8]. In their validation study [8], the original English version of the HEAS-13 demonstrated a theoretically meaningful multidimensional four-factor structure, including (1) *affective symptoms* (four items, e.g., “Feeling nervous, anxious or on edge”), (2) *rumination* (three items, e.g., “Unable to stop thinking about past events related to climate change”), (3) *behavioural symptoms* (three items, e.g., “Difficulty enjoying social situations with family and friends”), and (4) *anxiety about personal impact* (three items, e.g., “Feeling anxious about your personal responsibility to help address environmental problems”). These eco-anxiety indicators are assessed in terms of how much they have been experienced over the past two weeks, with responses provided on a four-point Likert scale ranging from 0 (“not at all”) to 3 (“nearly every day”) [8]. To date, the HEAS-13 has been validated in several different cultural populations, including Argentinian and Spanish [25], French [26], German [27], Italian [28], Portuguese [29], and Turkish [30,31] samples.

Existing studies have consistently indicated good psychometric performance of the HEAS-13 across different validation studies [32], with its empirically supported four-factor structure and good internal consistency reliability for subscale scores (Cronbach’s alpha) [8,25,26,27,28,29,30,31]. Previous works have indicated that the test–retest reliability of the HEAS-13 was moderate to good, with the following intraclass correlation coefficients (ICCs): from 0.38 to 0.59, at a twelve-week interval for the original English version [8]; from 0.50 to 0.64, at a two-week interval for the Argentinian–Spanish version [25]; from 0.47 to 0.56, at a three-week interval for the Turkish version [31]; and from 0.62 to 0.75 at a 7–10-day interval for the Portuguese version [29]. As for convergent and divergent validity, HEAS-13 scores were positively associated with engagement in pro-environmental behaviours [29], as well as stress, anxiety, and depression symptoms [8,27], supporting the idea that the assessment of eco-anxiety could be significant for the understanding of people’s psychological response to climate change.

### 1.3. Revealing the New Features of the Scale: Discriminant Validity and Potential Screening Properties

As a further contribution to the literature, assessing the discriminant validity of the HEAS-13 against people’s general psychological distress is highly pertinent. The first two items of the HEAS-13 affective symptom subscale (items 1 and 2) represent the two exact statements derived from the two-item Generalised Anxiety Disorder Scale (GAD-2), which is a widely used screening measure of anxiety symptoms [33]. Despite the fact that the HEAS-13’s instructions ask about climate-related worries (compared to the GAD-2’s instructions asking about worries in general), a potential overlap between the affective symptom subscale and the GAD-2 seems to be possible. Due to this, the question of whether eco-anxiety was statistically independent of the general symptoms of anxiety and depression experienced by people in their daily lives should be addressed.

The issue of a possible overlap is worth investigating and is highly clinically relevant, as some authors have questioned whether eco-anxiety is a unique and distinct construct (see, for review, [8,34]). From the theoretical and practical point of view, examining this issue could enhance our understanding of eco-anxiety and its links to affective outcomes (see, for instance, a comment by Sampaio and Sequeira [35] and a reply to it by Bhullar et al. [36]). If, in fact, the HEAS-13 specifically measures eco-anxiety (rather than general psychological distress or mental health disorder symptoms), researchers can be sure that eco-anxiety levels could act as unique correlates of different variables and that the predictive role of eco-anxiety is not simply a proxy for a general psychopathology.

Due to the above-indicated fact that the two items of the HEAS-13 affective symptom subscale reflect two exact statements derived from the GAD-2 and that the GAD-2 serves as a screening tool for general anxiety symptoms, we believe that the total score of these two HEAS-13 statements could be used as a screening measure of climate anxiety [33]. According to the GAD-2, a cut-off score ≥3 indicates the presence of a possible anxiety disorder [33]; thus, the same cut-off score could be used for these two HEAS-13 items. As this simple screening solution provides a percentage of people with clinically significant levels of eco-anxiety (i.e., individuals with a score ≥3), we feel that the proposed solution could be beneficial in research and practice, especially in cross-cultural studies.

### 1.4. The Current Study

In this study, we aimed to introduce the first Polish version of the HEAS-13 and examine its psychometric properties. Based on theory and the above-described research, we predicted that a four-factor (subscale) structure would perform well in our confirmatory factor analysis, with all subscales characterised by high levels of internal consistency reliability for our Polish HEAS-13. We were interested in examining the temporal stability of the scale within a two-week interval.

We anticipated that eco-anxiety would positively correlate with environmental concerns (i.e., egoistic, altruistic, and biospheric concerns), experience of climate change (i.e., a subjective level of perception of being affected by climate change), pro-environmental behaviour, and climate change worry. We also aimed to examine potential correlates of HEAS-13 scores among negative (i.e., anxiety and depression symptoms) and positive (i.e., well-being) mental health outcomes, predicting that eco-anxiety scores would be positively associated with psychopathology symptoms and negatively with well-being. We also investigated the discriminant validity of the HEAS-13 against people’s general psychological distress and demonstrated the implementation of a proposed screening feature of the scale.

## 2. Materials and Methods

### 2.1. Procedure

Our study was conducted in accordance with the Declaration of Helsinki’s Ethical Principles. The Maria Grzegorzewska University Ethics Committee approved the current study (No. 166/2024).

In this study, we used two data collection methods across two sample collection processes: online data collection (a purposeful sampling method with a maximum-variation design [37]) and a paper-and-pencil method. For our factor analytic study based on a sample from the general population in Poland, we invited Polish-speaking adults aged 18 or above to complete an online anonymous questionnaire form on a voluntary basis. Our survey was hosted on the Google Forms platform. From January to June 2024, we posted a link with an invitation and an appended consent form on Facebook and Instagram. Participants who provided their written informed consent digitally were able to complete the survey. A subset of this dataset has been used in a previous research paper [38]; however, it focused on different aims and hypotheses and does not have overlapping analyses.

Paper-and-pencil administrations were used for the test–retest examination of the Polish version of the HEAS-13. Our participants were social science students recruited at a Polish university during classes, where they completed the target questionnaire twice, at a two-week interval. This part of the study was also anonymous and voluntary. A unique participant code was used to match the same participants within these two time measurements. Because this sample was composed of university students and as we intended to examine a factor structure of the HEAS-13 in a sample from the general population in Poland, we did not include the data of participants obtained within this test–retest examination into the factor analytic part of this study.

### 2.2. Participants

Our main study sample consisted of 634 Polish-speaking adults (516 female, 110 male, and 8 non-binary participants) recruited from the general population in Poland, with ages ranging from 18 to 67 years, with a mean age of 28.12 (*SD* = 10.73). Table 1 indicates the detailed demographic characteristics of our sample. This sample was used in all our analyses, except for the test–retest examination.

Our test–retest study sample consisted of 95 Polish-speaking adults (82 female, 12 male, and 1 non-binary participants), with ages ranging from 18 to 57 years, with a mean age of 23.15 (*SD* = 7.59). As indicated above, this was the university sample composed of students of social sciences.

### 2.3. Measures

#### 2.3.1. The Demographic Questionnaire

Our participants filled out a sociodemographic form (with age, gender, education, and area of residence categories) and a series of psychometric self-report questionnaires, including our target measure of eco-anxiety (the HEAS-13), ecological variables (i.e., environmental concerns, experience of climate change, pro-environmental behaviour, and climate change worry), negative mental health outcomes (i.e., anxiety and depression symptoms), and positive mental health indicators (i.e., well-being). Internal consistency reliability coefficients for all the below-described questionnaires are displayed in the Results Section.

In our factor analytic study, a total of 634 participants completed all measures without missing data, except the measure on environmental concerns, which was filled out by 576 people. In our test–retest examination sample (*n* = 95), there were no missing data.

#### 2.3.2. The Hogg Eco-Anxiety Scale (HEAS-13)

The HEAS-13, developed by Hogg et al. [8], is a self-report questionnaire for assessing anxiety related to the climate and environmental crises. The scale measures four sub-dimensions of eco-anxiety: (1) affective symptoms (items 1, 2, 3, and 4), (2) rumination (items 5, 6, and 7), (3) behavioural symptoms (items 8, 9, and 10), and (4) anxiety about one’s negative impact on the planet (items 11, 12, and 13). The HEAS-13 consists of thirteen items (e.g., “Not being able to stop or control worrying”; “Feeling anxious about the impact of your personal behaviours on the earth”), with a four-point Likert scale (0 = not at all, 3 = nearly every day). Higher scores, calculated by averaging the items of the individual subscales, indicate higher levels of each of the eco-anxiety dimensions.

Our Polish translation of the HEAS-13 was prepared according to standard translation procedures [39]. First, the original version of the HEAS-13 was translated into Polish by five independent translators. Based on these translations, a common Polish translation was developed. Secondly, this translation was then back-translated into English by three independent translators. During these procedures, minor changes were made, resulting in the prefinal version of the Polish HEAS-13, which was then tested by ten Polish-speaking people from the general population. After their endorsement, it was regarded as the final version of the measure and administered in the present study. A copy of the measure can be found in the Appendix A.

#### 2.3.3. The Climate Change Worry Scale (CCWS)

The CCWS, developed by Stewart [24], is a self-report questionnaire for assessing the level of disturbing thoughts that people experience due to climate change. The scale consists of ten items (e.g., “Thoughts about climate change cause me to have worries about what the future may hold”; “I worry that outbreaks of severe weather may be the result of a changing climate”), with a five-point Likert scale (1 = never, 5 = always). A higher score, calculated by summing all the items, indicates a higher level of experienced climate change worry. The Polish version of this scale was developed by Larionow et al. [38].

#### 2.3.4. The Patient Health Questionnaire-4 (PHQ-4)

The PHQ-4, developed by Kroenke et al. [40], is a brief measure for assessing anxiety and depressive symptoms experienced in the previous two weeks. The questionnaire consists of two two-item subscales: (1) anxiety (e.g., “Feeling nervous, anxious or on edge”) and (2) depression (e.g., “Little interest or pleasure in doing things”). The PHQ-4 uses a four-point Likert scale (0 = not at all, 3 = nearly every day). The subscale scores can be calculated by summing the scores of the two anxiety items and two depression items, and a total score can be calculated by adding together the scores of each of the four items. The Polish version of the questionnaire was developed by Larionow and Mudło-Głagolska [41]. As this questionnaire was successfully used in previous climate-related Polish studies (e.g., [42]), we felt that the PHQ-4 was an appropriate tool for examining mental health correlates of eco-anxiety. Being a popular ultra-brief screening tool for anxiety and depression, the PHQ-4 was appropriate as just one of many constructs of interest in this study.

#### 2.3.5. The World Health Organisation—Five Well-Being Index (WHO-5)

The WHO-5, developed by The World Health Organization [43], is a short self-report questionnaire for measuring an individual’s current mental well-being. The WHO-5 consists of five items (e.g., “I have felt active and vigorous”; “I woke up feeling fresh and rested”), with a six-point Likert scale (0 = at no time, 5 = all of the time). The raw score can be calculated by totalling the figures of the five answers, and it ranges from 0 to 25, with 0 representing the worst possible well-being and 25 representing the best possible well-being. The Polish version of the WHO-5 was validated by Larionow [44].

#### 2.3.6. The Environmental Concern Scale (ECS)

The ECS, developed by Schultz [22], is a self-report questionnaire for measuring three areas of concern about environmental problems caused by human behaviours, with three four-item subscales: (1) egoistic concerns (e.g., “I am concerned about environmental problems because of the consequences for my health”), (2) altruistic concerns (e.g., I am concerned about environmental problems because of the consequences for people in my country”), and (3) biospheric concerns (e.g., “I am concerned about environmental problems because of the consequences for plants”). The scale consists of twelve items, with a seven-point Likert scale (1 = not important, 7 = supreme importance). Higher scores, calculated by averaging the items of the individual subscales, indicate higher levels of measured environmental concerns. The Polish version of the ECS was used previously in Polish studies [42].

#### 2.3.7. The Experience of Climate Change Scale (ECCS)

The ECCS, developed by Clayton and Karazsia [7], is a brief measure for assessing an individual’s perception of being affected by climate change. The ECCS consists of three items (e.g., “I have been directly affected by climate change”; “I have noticed a change in a place that is important to me due to climate change”), with a five-point Likert scale (1 = strongly disagree, 5 = strongly agree). A higher score, calculated by summing all the items, indicates a higher level of experience of climate change. The Polish version of the ECCS was used previously in Polish studies [38,42].

#### 2.3.8. The Behavioral Engagement Scale (BES)

The BES, developed by Clayton and Karazsia [7], is a brief measure of behavioural practices within environmental conservation efforts. The BES consists of six items (e.g., “I try to reduce my behaviors that contribute to climate change”; “I turn off lights”), with a five-point Likert scale (1 = strongly disagree, 5 = strongly agree). A higher score, calculated by summing all the items, indicates a higher level of pro-environmental behaviour. The Polish version of the BES was used previously in Polish studies [38,42].

### 2.4. Analytic Strategy

Statistical analyses were carried out using *Statistica* v. 13.3, *JASP* v. 0.19.0.0, and *R* v. 4.4.0 with the *lavaan* v. 0.6–18 statistical package.

#### 2.4.1. Descriptive Statistics and Demographic Differences

We computed means and standard deviations for all the study variables. Skewness and kurtosis scores were calculated for all individual HEAS-13 items. We compared the four HEAS-13 subscale scores between female and male participants with a frequentist independent sample *t*-test (with Cohen’s *d* as an effect size measure, with its interpretation based on the guidelines [45]), with the supplementation of a Bayesian independent sample *t*-test [46,47]. Spearman correlations between HEAS-13 subscale scores and demographic variables (i.e., age, education, and area of residence) were computed.

#### 2.4.2. Factor Structure

A confirmatory factor analysis with maximum-likelihood estimation (robust standard errors and the Satorra–Bentler scaled test statistic) was used. According to the statistical literature [48,49], this estimation method is one of the most preferable approaches for handling extreme non-normality. Absolute values of skewness >3 indicate that the distribution is extremely skewed [49].

We examined (1) a one-factor model, (2) the original and intended four-factor correlated model of the HEAS-13, (3) a second-order model with four first-order factors loaded on it, and (4) a bifactor model with an additional general eco-anxiety factor loading on all items, with the four factors (subscales) uncorrelated. The comparative fit index (CFI), the Tucker–Lewis index (TLI), the root mean square error of approximation (RMSEA), and the standardised root mean square residual (SRMR) were used as common fit indexes to evaluate model goodness-of-fit. CFI and TLI values ≥ 0.90 indicated acceptable fit and values ≥ 0.95 excellent fit. RMSEA and SRMR values ≤ 0.08 indicated acceptable fit and values ≤ 0.06 excellent fit [50].

#### 2.4.3. Internal Consistency and Test–Retest Reliability

McDonald’s omega (ω) and Cronbach’s alpha (α) reliability coefficients were calculated. Values ≥0.70 were judged as acceptable, ≥0.80 as good, and ≥0.90 as excellent [51]. For evaluating the test–retest reliability, we computed ICCs with their 95% confidence intervals (CIs). The form of ICC used was ICC_2,1_ (with two-way random effects, absolute agreement, and single rater/measurement) [52]. The ICC of HEAS-13 subscale scores between two time points with a two-week interval were calculated, with ICCs of <0.5 indicating poor reliability, ICCs between 0.5 and 0.75 indicating moderate reliability, ICCs between 0.75 and 0.9 indicating good reliability, and ICCs of >0.90 indicating excellent reliability [52]. Additionally, paired *t*-tests with their effect size measure (i.e., Cohen’s *d*) were used to compare HEAS-13 subscale scores between the two-time points. These tests were also supplemented with Bayesian paired *t*-tests [46].

#### 2.4.4. Concurrent Validity

For assessing concurrent validity, we calculated the Pearson correlations between the HEAS-13 scores and the other administered measures.

#### 2.4.5. Discriminant Validity

We were interested in assessing the discriminant validity of the HEAS-13 against general psychological distress at the *subscale* level, and, therefore, we conducted an exploratory factor analysis (based on parallel analysis) with all four HEAS-13 subscale scores and two PHQ-4 subscale scores (i.e., anxiety and depression symptoms). We predicted that the HEAS-13 subscale scores would not load on the same factor/factors composed of anxiety and depression symptoms, thus demonstrating the statistical separability of the eco-anxiety construct from general mental health symptoms.

Moreover, we examined the discriminant validity of the HEAS-13 against general psychological distress at the *item* level, using the same analysis as presented above for the subscale level, conducting it at item level. We put our measures’ items (i.e., all HEAS-13 and PHQ-4 items) in said analysis, predicting that the HEAS-13 items and the items of psychological distress would not cross-load, thus distinguishing the eco-anxiety construct from general distress.

#### 2.4.6. Screening Properties

In this study, we proposed a specific cut-off score for detecting clinically significant levels of eco-anxiety on the HEAS-13, with a score ≥3 of the first two HEAS-13 items indicating clinically significant levels of eco-anxiety.

## 3. Results

### 3.1. Descriptive Statistics and Demographic Differences

Table 2 presents descriptive statistics for all study variables in the total sample (*n* = 634) and different gender groups. In terms of the HEAS-13 scores, no statistically significant differences between these gender groups were noted for affective symptoms (*p* = 0.055, Cohen’s *d* = 0.20; BF_10_ = 0.69), rumination (*p* = 0.167, Cohen’s *d* = 0.15; BF_10_ = 0.29), and behavioural symptoms (*p* = 0.398, Cohen’s *d* = −0.09; BF_10_ = 0.16). Female participants tended to have statistically higher levels of anxiety about their personal impact than male participants (*p* < 0.001, Cohen’s *d* = 0.39; BF_10_ = 93.40).

In the total sample (*n* = 634), our Spearman correlational analysis indicated that age was statistically significantly and slightly negatively associated with HEAS-13 affective symptoms (*r* = −0.15, *p* < 0.001), rumination (*r* = −0.10, *p* < 0.05), behavioural symptoms (*r* = −0.10, *p* < 0.05), and anxiety about personal impact (*r* = −0.20, *p* < 0.001), suggesting that younger people experienced higher levels of eco-anxiety than older ones. Education was statistically significantly and slightly negatively related to affective symptoms (*r* = −0.12, *p* < 0.01), rumination (*r* = −0.09, *p* < 0.05), behavioural symptoms (*r* = −0.07, *p* > 0.05), and anxiety about personal impact (*r* = −0.16, *p* < 0.001), indicating that more educated people tended to have lower levels of eco-anxiety in most domains (except the statistically insignificant relationship with behavioural symptoms). Area of residence was not statistically significantly associated with HEAS-13 subscale scores.

### 3.2. Factor Structure

The values of skewness of all HEAS-13 items indicated that their distribution was not extremely skewed (see Appendix A). In the total sample (*n* = 634), the intended four-factor HEAS-13 model was a good fit for the data (χ^2^(59) = 156.393, CFI = 0.961, TLI = 0.949, RMSEA = 0.064 (90% CI [0.052, 0.076]), and SRMR = 0.058), with all standardised factor loadings being high on their intended factor (loadings ≥ 0.63, all *ps* < 0.001; see Appendix A). The estimated correlations between the HEAS-13 subscales are presented in Appendix A. These subscales were significantly positively correlated with each other, with estimated correlations from 0.17 to 0.69 (all *ps* < 0.001). In contrast, the one-factor model (χ^2^(65) = 967.213, CFI = 0.562, TLI = 0.475, RMSEA = 0.205 (90% CI [0.194, 0.217]), and SRMR = 0.149), the second-order model with the four first-order factors (χ^2^(61) = 260.368, CFI = 0.918, TLI = 0.896, RMSEA = 0.092 (90% CI [0.080, 0.103]), and SRMR = 0.111), and the bifactor model (χ^2^(52) = 208.874, CFI = 0.937, TLI = 0.905, RMSEA = 0.087 (90% CI [0.075, 0.100]), and SRMR = 0.096) were inferior to the intended four-factor model. No modifications to the models were provided.

### 3.3. Internal Consistency and Test–Retest Reliability

As demonstrated in Table 2, all HEAS-13 subscale scores had good internal consistency reliability, with ω and α of ≥0.79. Our test–retest results are reported in Appendix A. The ICCs of the HEAS-13 subscale scores between the two-time points were as affective symptoms (ICC = 0.72), rumination (ICC = 0.54), behavioural symptoms (ICC = 0.49), and anxiety about personal impact (ICC = 0.69), suggesting generally moderate test–retest reliability for the HEAS-13. We supplemented our ICCs with paired *t*-tests, and we revealed that the scores of affective symptoms and behavioural symptoms were statistically significantly different between the two-time points (with small effect sizes), whereas the scores of rumination and anxiety about personal impact were not statistically significantly different.

Our Bayesian paired *t*-tests provided additional information regarding the temporal stability of HEAS-13 scores. Based on these tests, we revealed substantial evidence in favour of H_1_ (BF_10_ = 9.27), i.e., that the mean scores of affective symptoms were different between the first and the last time points of a two-week interval, suggesting that affective symptoms seem to be the least stable dimension of eco-anxiety. In contrast, we revealed substantial evidence in favour of H_0_ (BF_10_ = 0.26), i.e., that there were no differences between the mean scores of anxiety about personal impact between the first and the last time points. In general, our results suggested moderate temporal stability of the Polish HEAS-13, with affective symptoms being the least temporally stable dimension of the eco-anxiety construct and with anxiety about personal impact being the most temporally stable one.

### 3.4. Concurrent Validity

Our Pearson correlations (Table 3) revealed that most HEAS-13 subscale scores (except for behavioural symptoms) were statistically significantly and positively associated with egoistic concerns, altruistic concerns, biospheric concerns, experience of climate change, and pro-environmental behavioural engagement. All HEAS-13 subscale scores were statistically significantly and positively related to climate change worry and anxiety and depression symptoms. In contrast, all HEAS-13 subscale scores were statistically significantly and negatively related to well-being. All these results suggested good convergent and divergent validity of the Polish HEAS-13.

### 3.5. Discriminant Validity

In the total sample (*n* = 634), exploratory factor analysis of the four HEAS-13 subscales and the two PHQ-4 subscales extracted three meaningful factors (see Table 4). The two PHQ-4 subscales were combined into the general psychological distress factor (Factor 1), whereas HEAS-13 rumination and anxiety about personal impact and HEAS-13 affective and behavioural symptoms were components of Factor 2 and Factor 3, respectively. As expected, each factor had salient factor loadings, with no cross-loadings. As such, the HEAS-13 subscale scores showed good discriminant validity against general psychological distress.

In the total sample (*n* = 634), exploratory factor analysis of HEAS-13 items and PHQ-4 items extracted five meaningful factors (see Table 5). The four PHQ-4 items combined into a general psychological distress factor (Factor 1), whereas the other four factors (Factors 2–5) were the four subscales of the HEAS-13, with their intended items. As expected, each factor had salient factor loadings, with no cross-loadings. As such, the HEAS-13 items showed good discriminant validity against the PHQ-4 items.

### 3.6. Screening Properties

In this paper, we proposed a new screening feature of the HEAS-13, that is, a total sum score of the first two HEAS-13 items ≥3 potentially indicating a clinically significant level of eco-anxiety. Using this cut-off, in the total sample (*n* = 634), 80 people (12.62% of the total sample) had clinically significant levels of eco-anxiety.

We also provided this analysis in regard to gender, leading to the following observations. In the total sample of female participants (*n* = 516), 66 (12.79% of the total female sample) had clinically significant levels of eco-anxiety. In the total sample of male participants (*n* = 110), 13 (11.82% of the total male sample) had clinically significant levels of eco-anxiety. In the total sample of non-binary people (*n* = 8), one individual (12.50% of the total non-binary sample) had a clinically significant level of eco-anxiety.

## 4. Discussion

In this study, we examined the psychometric properties of the first Polish version of the HEAS-13 for measuring eco-anxiety. Overall, the HEAS-13 demonstrated good psychometric performance, thus supporting the notion that it provides a strong measure of the multidimensional eco-anxiety construct.

### 4.1. Factor Structure and Internal Consistency Reliability

As expected, the Polish HEAS-13 was characterised by an intended four-factor solution, with all subscale scores having good internal consistency. These results are in line with the large body of previous validation studies across different countries [25,26,27,28,29,30,31], including the original development study [8], suggesting that the HEAS-13 seems to have good cross-cultural applicability. We also demonstrated that the one-factor model, the second-order model with the four first-order factors, and the bifactor model were inferior to the intended four-factor correlated model. This indicates that the four HEAS-13 subscale scores are best used at the subscale level and should not be summed into an overall score of eco-anxiety.

### 4.2. Temporal Stability

In our study, we examined the temporal stability of HEAS-13 scores within a two-week interval using a series of different methodological approaches, including ICCs as well as frequentist and Bayesian paired *t*-tests. Based on all these analyses, we found that anxiety about personal impact was the most temporally stable dimension of eco-anxiety, followed by rumination, behavioural symptoms, and then affective symptoms. This conclusion supports the results presented in the original validation study of the HEAS-13 [8], indicating that different eco-anxiety dimensions could be more or less persistent over time. Overall, the eco-anxiety construct, as measured by the HEAS-13, seems to be relatively stable over time. This insight could inform the timing and targeting of interventions aimed at mitigating distressing levels of eco-anxiety.

### 4.3. Concurrent Validity

Among the correlates of eco-anxiety, we examined climate-related variables (i.e., environmental concerns, experience of climate change, pro-environmental behaviours, and climate change worry) and clinically relevant variables on negative mental health (i.e., anxiety and depression symptoms) and positive mental health (i.e., well-being). Our correlational analysis demonstrated that higher levels of eco-anxiety were associated with higher levels of egoistic concerns, altruistic concerns, and, especially, biospheric concerns. This indicates that eco-anxiety, as measured by the HEAS-13, is associated with environmental worries due to negative impacts on a range of areas. As expected, HEAS-13 scores were positively associated with the experience of climate change (i.e., a subjective level of awareness of being affected by climate change) and pro-environmental behaviours in our sample, suggesting that moderate levels of eco-anxiety could stimulate, maintain, or direct pro-environmental efforts [53]. We also noted moderate-to-strong positive correlations between HEAS-13 scores (especially anxiety about personal impact and rumination scores) and climate change worry, illustrating their theoretical affinity.

As expected, all HEAS-13 subscale scores were positively associated with anxiety and depression symptoms and, negatively, with well-being, suggesting links between eco-anxiety and general mental health issues. While previous studies on the HEAS-13 mainly focused on ill-being indicators (i.e., anxiety symptoms, stress) as correlates of eco-anxiety, we supplemented our analyses with a well-being indicator. Higher levels of eco-anxiety (especially affective symptoms and behavioural symptoms) were associated with lower levels of well-being, emphasising potential negative effects of eco-anxiety symptoms on people’s overall well-being. Thus, eco-anxiety seems to be an important correlate of overall ill-being and well-being, supporting previous evidence in this regard [27,30]. It is essential to emphasise that the examined associations are based on our cross-sectional study; therefore, we cannot determine causality/directionality here. As we treat these associations as bi-directional, it can be suggested that people more predisposed to general anxiety will manifest it in many domains, including climate areas. At the same time, people’s climate anxiety can contribute to an overall poorer well-being, which, in turn, could trigger general anxiety tendencies.

Overall, the results emphasise the importance of addressing distressing climate-related fears in mental health interventions. However, the findings also suggest that eco-anxiety can trigger positive pro-environmental behaviour through constructive environmental engagement. Understanding this dual nature of eco-anxiety can help researchers and practitioners develop strategies which will leverage individuals’ emotional responses to climate change and foster community involvement.

### 4.4. Discriminant Validity

In this paper, one of our core analyses was also devoted to answering the question of whether the eco-anxiety construct, as measured by the HEAS-13, was statistically separable from general anxiety and depression symptoms. Our data evidenced that eco-anxiety and general psychological distress were separable constructs, suggesting that the HEAS-13 measure of eco-anxiety can evaluate a unique eco-anxiety construct.

Overall, our psychometric evidence suggests that the HEAS-13 seems to be a highly valid and reliable measure. Its strong psychometric properties demonstrate that eco-anxiety can be effectively measured across different cultural contexts. This supports the cross-cultural relevance of anxiety toward climate-related issues.

### 4.5. Demographic Comparisons

As previous studies evidenced gender differences in HEAS-13 scores (i.e., female participants had higher levels of eco-anxiety scores than male participants) [25], we were interested in examining this in our study as well. In line with the past work [25], we noted that female participants had statistically higher levels of anxiety about personal impact than males; however, no statistically significant gender differences were present in other aspects of eco-anxiety. Our younger participants experienced higher levels of climate-related anxiety than our older ones, further supporting existing evidence [25,42,54,55,56]. In line with previous Polish studies on climate anxiety [38,42], we noted that more educated people had lower levels of eco-anxiety, whereas people’s area of residence was not statistically significantly associated with eco-anxiety. These results indicate that it may be necessary to control for education levels in statistical analyses in future ecopsychological studies (e.g., [10]). The finding that more educated individuals tend to have lower levels of eco-anxiety suggests that education may influence how individuals perceive and cope with climate-related issues. This implication points to the necessity of promoting climate education to potentially help alleviate distressing levels of eco-anxiety.

As our sample was not balanced regarding different demographic categories, we feel that these demographic comparisons should be considered tentative. Future studies should take into account potential demographic differences in eco-anxiety levels. Studying these demographic differences has implications for targeting specific populations in both research and interventions related to eco-anxiety.

### 4.6. Screening for Elevated Eco-Anxiety

In this study, we proposed a potential screening feature of the HEAS-13 and showed its implementation. To the best of the authors’ knowledge, this feature has not been demonstrated previously. Overall, our results indicated that about one out of eight people, regardless of gender, tended to have a clinically significant level of eco-anxiety (i.e., a total sum scores of the first two HEAS-13 items ≥3). We feel that this screening feature may have strong potential in cross-cultural research, as it provides an easy metric to capture clinically significant levels of eco-anxiety and compare proportions of people across populations. For some uses, these percentage levels may be more intuitive than, for instance, mean scores with their standard deviations. Future research into its use will be beneficial.

### 4.7. Practical Implications of This Study

The Polish version of the HEAS-13 demonstrated good psychometric properties and, therefore, can be recommended for application in studies investigating eco-anxiety levels. For instance, researchers can explore the subjective perception of eco-anxiety across different demographics (e.g., age and gender groups, as well as socio-economic backgrounds). The screening feature of the scale may be useful in cross-cultural studies.

A growing body of literature demonstrates that the most vulnerable groups to eco-anxiety and its consequences include children and young adults and individuals directly impacted by environmental disasters [57]. Consequently, given the adverse mental health effects resulting from worries about climate change [58], it is important to conduct studies on these populations worldwide. This global perspective can help inform tailored interventions and support systems, ensuring that the unique needs of these at-risk groups are effectively addressed.

### 4.8. Limitations of This Study and Future Directions

Whilst our study represents a good contribution to the field, it has several limitations that can be built on in future work. First, this was a cross-sectional study, thus limiting determinations of the cause-and-effect relationship between eco-anxiety and its correlates. Therefore, future work should be based on longitudinal analyses, to determine the predictive ability of eco-anxiety in mental health outcomes and pro-environmental behaviours.

Secondly, we examined the psychometrics of the Polish HEAS-13 in a sample from the general population, and our sample was mostly young people and female participants with secondary education and from large cities (above 100,000 inhabitants), so future work in more diverse samples, including clinical ones, will be beneficial to test the generalizability of our findings. To mitigate the limitation of the self-selection bias in the participant recruitment process, the purposeful sampling method with a maximum-variation design [37] and the diversification of social networking sites during recruitment advertising were used.

Thirdly, our test–retest sample comprised students, with a limitation of an imbalanced gender ratio. However, our test–retest sample was large; therefore, we feel that our analyses with the frequentist and Bayesian test–retest examination represent a good contribution to the field. Future work could examine generalizability to other population types.

Finally, this was principally an online study (except for the test–retest procedure); therefore, it would be beneficial to collect data using a paper-and-pencil method in future studies and directly compare the psychometrics of the online and paper-and-pencil forms of the HEAS-13.

## 5. Conclusions

The Polish version of the HEAS-13 is a valid and reliable measure of the multidimensional eco-anxiety construct. Much like the other language versions, the Polish HEAS-13 is characterised by an intended four-factor structure and has good internal consistency and moderate test–retest reliability, as well as good concurrent validity. The eco-anxiety construct seems to be distinguishable from general psychological distress. Our findings highlight the importance of addressing eco-anxiety as a significant psychological phenomenon. This validation of the HEAS-13 makes further theoretical and empirical examination of the eco-anxiety construct possible.

## Figures and Tables

**Table 1 healthcare-12-02255-t001:** Demographic characteristics of the study sample.

Demographic Characteristics		*n*	%
Age	*M* = 28.12, *SD* = 10.73, median = 22.00, min. = 18, max. = 67	634	100
Gender	Female	516	81.39
	Male	110	17.35
	Non-binary	8	1.26
Education	Higher	242	38.17
	Secondary	384	60.57
	Vocational	4	0.63
	Primary	4	0.63
Area of residence	Large cities (above 100,000 inhabitants)	287	45.27
	Towns (from 20,000 to 100,000)	103	16.25
	Small towns (up to 20,000)	89	14.04
	Villages	155	24.45

**Table 2 healthcare-12-02255-t002:** Descriptive statistics and internal consistency reliability coefficients for the study variables.

Scale/Subscale	Total Sample					Female			Male			Non-Binary		
	*n*	ω (95% CI)	α (95% CI)	*M*	*SD*	*n*	*M*	*SD*	*n*	*M*	*SD*	*n*	*M*	*SD*
HEAS-13 Affective symptoms	634	0.84 (0.82; 0.86)	0.84 (0.82; 0.86)	0.47	0.58	516	0.49	0.58	110	0.38	0.61	8	0.56	0.48
HEAS-13 Rumination	634	0.85 (0.83; 0.87)	0.85 (0.82; 0.86)	0.37	0.51	516	0.38	0.52	110	0.31	0.47	8	0.42	0.50
HEAS-13 Behavioural symptoms	634	0.80 (0.78; 0.83)	0.79 (0.76; 0.82)	0.34	0.58	516	0.33	0.56	110	0.38	0.68	8	0.38	0.45
HEAS-13 Anxiety about personal impact	634	0.84 (0.82; 0.86)	0.84 (0.81; 0.86)	0.47	0.58	516	0.51	0.60	110	0.28	0.40	8	0.63	0.49
ECS Egoistic concerns	576	0.92 (0.91; 0.93)	0.92 (0.91; 0.93)	4.81	1.57	469	4.95	1.50	103	4.13	1.72	4	5.44	0.94
ECS Altruistic concerns	576	0.91 (0.90; 0.92)	0.91 (0.90; 0.92)	5.12	1.57	469	5.26	1.51	103	4.45	1.69	4	6.00	0.74
ECS Biospheric concerns	576	0.94 (0.93; 0.95)	0.94 (0.93; 0.95)	4.99	1.63	469	5.12	1.59	103	4.40	1.71	4	6.13	0.48
ECCS Experience of climate change	634	0.79 (0.76; 0.82)	0.79 (0.76; 0.82)	10.28	3.19	516	10.42	3.09	110	9.42	3.58	8	12.75	1.04
BES Pro-environmental behavioural engagement	634	0.78 (0.75; 0.80)	0.77 (0.74; 0.80)	24.76	4.27	516	25.23	3.97	110	22.48	4.97	8	25.75	2.38
CCWS Climate change worry	634	0.93 (0.93; 0.94)	0.93 (0.92; 0.94)	24.64	8.35	516	25.34	8.30	110	20.88	7.59	8	31.13	5.06
PHQ-4 Anxiety	634	0.85 (0.82; 0.87)	0.85 (0.82; 0.87)	2.15	1.72	516	2.25	1.73	110	1.58	1.54	8	3.50	1.51
PHQ-4 Depression	634	0.84 (0.81; 0.87)	0.84 (0.82; 0.87)	1.95	1.80	516	1.98	1.81	110	1.76	1.72	8	2.50	2.20
PHQ-4 Total score	634	0.88 (0.87; 0.90)	0.88 (0.87; 0.90)	4.10	3.25	516	4.23	3.28	110	3.35	3.03	8	6.00	2.98
WHO-5 Well-being	634	0.87 (0.85; 0.89)	0.87 (0.86; 0.89)	10.93	4.89	516	10.87	4.88	110	11.30	5.05	8	9.88	2.10

**Table 3 healthcare-12-02255-t003:** Pearson correlations between the HEAS-13 subscale scores and other study variables.

Variables	HEAS-13 Affective Symptoms	HEAS-13 Rumination	HEAS-13 Behavioural Symptoms	HEAS-13 Anxiety About Personal Impact
ECS Egoistic concerns	0.13 **	0.30 ***	0.06	0.31 ***
ECS Altruistic concerns	0.16 ***	0.31 ***	0.04	0.32 ***
ECS Biospheric concerns	0.22 ***	0.38 ***	0.04	0.37 ***
ECCS Experience of climate change	0.17 ***	0.31 ***	0.05	0.35 ***
BES Pro-environmental behavioural engagement	0.16 ***	0.26 ***	0.04	0.31 ***
CCWS Climate change worry	0.38 ***	0.54 ***	0.14 ***	0.56 ***
PHQ-4 Anxiety	0.34 ***	0.14 ***	0.25 ***	0.22 ***
PHQ-4 Depression	0.33 ***	0.15 ***	0.33 ***	0.21 ***
PHQ-4 Total score	0.36 ***	0.16 ***	0.32 ***	0.24 ***
WHO-5 Well-being	−0.25 ***	−0.09 *	−0.28 ***	−0.15 ***

Note. Correlations between HEAS-13 subscale scores and ECS subscale scores are based on a sample of 576 people, whereas correlations between HEAS-13 subscale scores and all other scores are based on the total sample of 634 people. * *p* < 0.05; ** *p* < 0.01; and *** *p* < 0.001.

**Table 4 healthcare-12-02255-t004:** Factor loadings from the parallel analysis of the HEAS-13 and PHQ-4 subscale scores (*n* = 634).

Variables	Factor 1 “General Psychological Distress”	Factor 2 “Rumination and Anxiety About Personal Impact”	Factor 3 “Affective and Behavioural Symptoms of Eco-Anxiety”
HEAS-13 Affective symptoms			0.72
HEAS-13 Rumination		0.77	
HEAS-13 Behavioural symptoms			0.71
HEAS-13 Anxiety about personal impact		0.70	
PHQ-4 Anxiety	0.82		
PHQ-4 Depression	0.81		
The proportion of the variance explained by each rotated factor	23.69%	20.57%	19.02%

Note. The applied parallel analysis was based on the correlation matrix and factor eigenvalues. The factoring method was principal axis factoring, with varimax rotation. The cumulative proportion of the variance explained by the rotated factors was 63.27%. For clarity reasons, factor loadings below 0.40 are not shown.

**Table 5 healthcare-12-02255-t005:** Factor loadings from the parallel analysis of the HEAS-13 and PHQ-4 items (*n* = 634).

Variables	Factor 1 “General Psychological Distress”	Factor 2 “HEAS-13 Affective Symptoms”	Factor 3 “HEAS-13 Rumination”	Factor 4 “HEAS-13 Anxiety About Personal Impact”	Factor 5 “HEAS-13 Behavioural Symptoms”
HEAS-13 item 1		0.47			
HEAS-13 item 2		0.77			
HEAS-13 item 3		0.76			
HEAS-13 item 4		0.60			
HEAS-13 item 5			0.72		
HEAS-13 item 6			0.77		
HEAS-13 item 7			0.71		
HEAS-13 item 8					0.60
HEAS-13 item 9					0.71
HEAS-13 item 10					0.77
HEAS-13 item 11				0.69	
HEAS-13 item 12				0.83	
HEAS-13 item 13				0.70	
PHQ-4 Anxiety item 1	0.79				
PHQ-4 Anxiety item 2	0.80				
PHQ-4 Depression item 1	0.84				
PHQ-4 Depression item 2	0.72				
The proportion of the variance explained by each rotated factor	15.80%	12.63%	12.06%	11.86%	11.20%

Note. The applied parallel analysis was based on the correlation matrix and factor eigenvalues. The factoring method was principal axis factoring, with varimax rotation. The cumulative proportion of the variance explained by the rotated factors was 63.54%. For clarity reasons, factor loadings below 0.40 are not shown.

## Data Availability

The raw data supporting the conclusions of this article will be made available by the authors upon request.

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
