# Peer review of "Measuring Eco-Anxiety with the Polish Version of the 13-Item Hogg Eco-Anxiety Scale (HEAS-13): Latent Structure, Correlates, and Psychometric Performance"

_healthcare, 2024, doi:10.3390/healthcare12222255_

Round 1

Reviewer 1 Report

Comments and Suggestions for Authors
  1. Please change the title to "Latent Structure, Correlates, and Psychometric Performance of the Polish Version of the 13-Item Hogg Eco-Anxiety Scale (HEAS-13)."

  2. Verify that the keywords include five terms using Meshbrowser.

  3. The introduction's final paragraph should outline the study's objectives related to the methods and measurements.

  4. The notes beneath each table are too detailed; some information should be incorporated into the results section instead of included as notes.

  5. In the discussion section, the last sentence before addressing the study's limitations should highlight the clinical implications, such as assessing anxiety levels related to climate change or eco-influence in potential psychiatric cases and how to leverage policy to support environmental legislation.

  6. Please recommend additional studies based on the limitations identified in this research.

  7. Update the references to include works from 2014 to 2024, except classic models and measurement validity, such as Reference 49, which is from 1998.

Author Response

We would like to thank the editor and the reviewers for their very positive and encouraging feedback on our submission. The constructive comments of the reviewers helped us to significantly improve the quality of our submission. We have been through all comments one by one, edited the manuscript in detail, and added new material where required. We hope the editor and reviewers find the revised version of the manuscript clear and suitable for publication in Healthcare. All changes made in the manuscript are in red.

REVIEWER 1:

Please change the title to "Latent Structure, Correlates, and Psychometric Performance of the Polish Version of the 13-Item Hogg Eco-Anxiety Scale (HEAS-13)."

Response: We have now changed the title.

Verify that the keywords include five terms using Meshbrowser.

Response: We have verified it and we can conclude that our keywords include at least five MeSH terms.

The introduction's final paragraph should outline the study's objectives related to the methods and measurements.

Response: Our study's objectives related to the methods and measurements have now been outlined in the section The Current Study, where we explained our aim.

The notes beneath each table are too detailed; some information should be incorporated into the results section instead of included as notes.

Response: Thank you for this comment. We put some less important data into the notes in order to make the text easy to read. We would not like to change the flow and focus of the most relevant data into these less important details, therefore, we would like to keep our notes. However, we would be happy to further edit this if the editorial team would like to request this.

In the discussion section, the last sentence before addressing the study's limitations should highlight the clinical implications, such as assessing anxiety levels related to climate change or eco-influence in potential psychiatric cases and how to leverage policy to support environmental legislation.

Response: We have now elaborated on clinical implications, in terms of how the levels of eco-anxiety can be assessed with the scale in cross-cultural and clinical research. Also the use of a cut-off score to inform interpretation of meaningfully elevated scores.

Please recommend additional studies based on the limitations identified in this research.

Response: We have now provided ideas for additional studies.

Update the references to include works from 2014 to 2024, except classic models and measurement validity, such as Reference 49, which is from 1998.

Response: We have now provided the most relevant literature in our references.

Reviewer 2 Report

Comments and Suggestions for Authors

Thanks to the authors for sharing their manuscript. Appreciating the manuscript and the conducted research, I would like to mention a few points.

Introduction

Eco-anxiety is a very new and little-studied phenomenon, and therefore I would like to see a more detailed theoretical review of eco-anxiety (1.1. The Eco-Anxiety Construct).

Materials and methods

The authors describe the measures in sufficient detail and refer to the fact that their Polish versions were used, but do not cite the Cronbach's alpha coefficients for the study sample.

Results

It remains unclear why the authors limited themselves to evaluating the four-factor structure of the instrument. I think it was worth checking at least the bifactor structure (perhaps this is not necessary according to previous studies, but it can be mentioned).

These comments are of a recommendatory nature and can be taken into account at the request of the authors.

Sincerely yours,

the Reviewer.

Author Response

We would like to thank the editor and the reviewers for their very positive and encouraging feedback on our submission. The constructive comments of the reviewers helped us to significantly improve the quality of our submission. We have been through all comments one by one, edited the manuscript in detail, and added new material where required. We hope the editor and reviewers find the revised version of the manuscript clear and suitable for publication in Healthcare. All changes made in the manuscript are in red.

Thanks to the authors for sharing their manuscript. Appreciating the manuscript and the conducted research, I would like to mention a few points.

Introduction

Eco-anxiety is a very new and little-studied phenomenon, and therefore I would like to see a more detailed theoretical review of eco-anxiety (1.1. The Eco-Anxiety Construct).

Response: We have now added some more information about the eco-anxiety construct into the introduction section

Materials and methods

The authors describe the measures in sufficient detail and refer to the fact that their Polish versions were used, but do not cite the Cronbach's alpha coefficients for the study sample.

Response: Thank you for your comment. Internal consistency reliability coefficients were provided in our Table 2, with 95% CI. We usually provide internal consistency reliability coefficients in the results section as these coefficients are a part of results in a psychometrically focused paper. This is the approach we have taken here, outlining those values in the results rather than the methods section. To help reader clarity on this, in the revised manuscript we have now added a sentence in the measures section, which refers to these coefficients presented in the results.

Results

It remains unclear why the authors limited themselves to evaluating the four-factor structure of the instrument. I think it was worth checking at least the bifactor structure (perhaps this is not necessary according to previous studies, but it can be mentioned).

Response: We have now additionally tested a one-factor model, a higher-order model with the four factors, and a bi-factor model We have demonstrated that the one-factor model and the second-order model with the four first-order factors as well as the bi-factor model were inferior to the intended four-factor model. This indicates that the four HEAS-13 subscale score cannot be summed into an overall score of eco-anxiety. We think this additional model testing helps reveal more about the nature of the latent structure of the construct.

These comments are of a recommendatory nature and can be taken into account at the request of the authors.

Sincerely yours,

the Reviewer.

Reviewer 3 Report

Comments and Suggestions for Authors

Dear Authors,

The study is a potentially useful contribution to the field, and the results have the potential to be of significant value. Nevertheless, I would like to propose a few suggestions for your consideration and further elaboration.

Introduction

The paragraphs on lines 106-114 and 115-131 are in the section where only the specification of the aims of the current study is expected. However, the content of these two paragraphs goes significantly beyond the expected focus, making it difficult to read. Therefore, I recommend that the authors consider moving some of the information (especially general explanations of each topic, examples, and a review of previous findings) from these paragraphs to another location within the Introduction.

2.1 Procedure

The sample used in the first study is so disproportionate that it requires further comment and explanation. The current format is inadequate, although it is mentioned in the Discussion. This is not only about gender, but also about age and education (and most likely a number of other characteristics).

Please elaborate on which "community" was involved in this case? How was the theoretical population defined? How does the sample differ from this theoretical population?

What sampling frame was used, or how was the list of potential respondents constructed? I assume this was not an online panel, was it?

What was the relationship of the respondents to the topic or environmental issue? Community in this case could also mean a community of environmentalists. If it is a regionally defined community, it would be useful to explain the context in the form of a description of the external factors in the life of that community.

For the second sample (used for the test-retest study), please add similar information as appropriate.

2.3.4 PHQ-4

Please include an explanation of why this version of the test was used instead of the usual PHQ-7.

2.4 Analytical strategy

Please provide a clear statement of what analyses were performed on the first and second datasets (samples). Ideally, this information can be included in the notes to the relevant tables, but the authors will surely find their own appropriate way.

3. Results

Extreme skewness is evident from the results presented (as evidenced by the values in Supplementary Table 1 and further indicated by the values of the means in Table 2). Under these circumstances, the results of the factor analyses performed are unreliable. I recommend that the authors repeat the analyses and take appropriate measures (WLS, data transformation, polychoric correlations, etc.).

3.2 Factor structure

Please add information on whether alternative models were tested or how the model was optimized.

From the manuscript it seems that CFA and EFA were performed on the same data, which would not be correct. Therefore, I would ask for a clearer specification of the specific data samples used or a revision of the analyses performed.

Table 2

Please consider whether it is appropriate to provide confidence intervals when the sample used is clearly not representative and the values obtained cannot be generalized, i.e., there are no inferential claims.

4. Discussion

Please consider whether the information in the paragraph on lines 458-471 is misleading with respect to the characteristics of your sample (comparing the characteristics of your sample with those of the samples in other studies).

Include the limitations of the research conducted in the spirit of the comments above.

5. Conclusions

Please consider adding information about how the results can be used in academic settings and in practice.

I am grateful for your commitment to advancing the understanding of this topic. I eagerly await the opportunity to review the revised version of your paper.

Sincerely,

Author Response

We would like to thank the editor and the reviewers for their very positive and encouraging feedback on our submission. The constructive comments of the reviewers helped us to significantly improve the quality of our submission. We have been through all comments one by one, edited the manuscript in detail, and added new material where required. We hope the editor and reviewers find the revised version of the manuscript clear and suitable for publication in Healthcare. All changes made in the manuscript are in red.

REVIEWER 3:

Dear Authors,

The study is a potentially useful contribution to the field, and the results have the potential to be of significant value. Nevertheless, I would like to propose a few suggestions for your consideration and further elaboration.

Introduction

The paragraphs on lines 106-114 and 115-131 are in the section where only the specification of the aims of the current study is expected. However, the content of these two paragraphs goes significantly beyond the expected focus, making it difficult to read. Therefore, I recommend that the authors consider moving some of the information (especially general explanations of each topic, examples, and a review of previous findings) from these paragraphs to another location within the Introduction.

Response: We have now moved some of the information from these paragraphs to another location within the introduction.

2.1 Procedure

The sample used in the first study is so disproportionate that it requires further comment and explanation. The current format is inadequate, although it is mentioned in the Discussion. This is not only about gender, but also about age and education (and most likely a number of other characteristics).

Response: We have added this as a limitation.

Please elaborate on which "community" was involved in this case? How was the theoretical population defined? How does the sample differ from this theoretical population? What sampling frame was used, or how was the list of potential respondents constructed? I assume this was not an online panel, was it? What was the relationship of the respondents to the topic or environmental issue? Community in this case could also mean a community of environmentalists. If it is a regionally defined community, it would be useful to explain the context in the form of a description of the external factors in the life of that community.

Response: In the previous version of our paper, we referred to a "general community sample", or as "a sample from the general population in Poland". Due to some potential ambiguities of the word "community", we have now replaced it with "a sample from the general population in Poland". These are usual people from the general population, without any specific features that were targeted during recruitment. We understand that our sample is not representative, but it is relatively diverse, and this situation is common for the psychometric studies. As this was a self-report study, with the recruitment of participants possibly characterised by self-selection bias or other characteristics. To mitigate this limitation, diversifying social networking sites during the recruitment process was used. These were not sites with the environmental content. So, we feel that such practices mitigate the indicated limitations.

For the second sample (used for the test-retest study), please add similar information as appropriate.

Response: We have now added this information.

2.3.4 PHQ-4

Please include an explanation of why this version of the test was used instead of the usual PHQ-7.

Response: As this questionnaire was successfully used in previous climate-related Polish studies (e.g., [Larionow et al., 2022]), we felt the PHQ-4 was an appropriate tool for examining mental health correlates of eco-anxiety. Being a popular ultra-brief screening tool of anxiety and depression, the PHQ-4 was very appropriate as just one of many constructs of interest in this study.

2.4 Analytical strategy

Please provide a clear statement of what analyses were performed on the first and second datasets (samples). Ideally, this information can be included in the notes to the relevant tables, but the authors will surely find their own appropriate way.

Response: We have now clarified this. Each table and each analysis have now been supplemented with number of participants analyzed.

3. Results

Extreme skewness is evident from the results presented (as evidenced by the values in Supplementary Table 1 and further indicated by the values of the means in Table 2). Under these circumstances, the results of the factor analyses performed are unreliable. I recommend that the authors repeat the analyses and take appropriate measures (WLS, data transformation, polychoric correlations, etc.).

Response: We understand that different authors prefer different statistical analyses. We base our choice of robust ML estimator on much of the past statistical literature [Brown, 2006; Harrington, 2009], which indicates that this estimation method is one of the most preferable approaches for handling extreme non-normality (compared to other estimators, including WLS). The ML estimator (and its robust variants) is commonly used in different HEAS-13 studies, therefore, for comparability reasons and with consistency with other studies, we presented the results with robust ML estimator. The robust ML estimator does not assume normality in the data, and thus is robust for use with samples that present skewed distributions.

3.2 Factor structure

Please add information on whether alternative models were tested or how the model was optimized.

Response: We have now added this information into the paper. No modifications to the models were provided.

From the manuscript it seems that CFA and EFA were performed on the same data, which would not be correct. Therefore, I would ask for a clearer specification of the specific data samples used or a revision of the analyses performed.

Response: The factor structure of the HEAS-13 was tested using the CFA in the whole sample (n = 634). We did not use EFA to test the factor structure of the HEAS-13. The EFA was used specifically for testing discriminant validity, seeing whether HEAS-13 scores cross-loaded on other constructs; therefore, these analyses EFA and CFA are not overlapping. We have now specified the data samples and the analyses. Each table and each analyses have now been supplemented with number of participants analyzed.

Table 2

Please consider whether it is appropriate to provide confidence intervals when the sample used is clearly not representative and the values obtained cannot be generalized, i.e., there are no inferential claims.

Response: The reporting of confidence intervals is, to our understanding, recommended practice regardless of the sample type. We feel that this is appropriate as we provide confidence intervals for internal reliability coefficients as recommended in the literature: Iacobucci, D., & Duhachek, A. (2003). Advancing alpha: Measuring reliability with confidence. Journal of Consumer Psychology, 13(4), 478–487. https://doi.org/10.1207/S15327663JCP1304_14

4. Discussion

Please consider whether the information in the paragraph on lines 458-471 is misleading with respect to the characteristics of your sample (comparing the characteristics of your sample with those of the samples in other studies).

Response: We have now added some clarifications: "As our sample was not balanced regarding different demographic categories, we feel that these demographic comparisons should be considered tentative. Future studies should take into account potential demographic differences in eco-anxiety levels by controlling the demographic variables in statistical analyses".

Include the limitations of the research conducted in the spirit of the comments above.

Response: We have now added more limitations based on your comments.

5. Conclusions

Please consider adding information about how the results can be used in academic settings and in practice.

Response: We have now created a practical implications section, where we described potential implications of the study in more detail.

I am grateful for your commitment to advancing the understanding of this topic. I eagerly await the opportunity to review the revised version of your paper.

Sincerely,

Once again, thank you for your helpful comments.

Round 2

Reviewer 3 Report

Comments and Suggestions for Authors

Dear Authors,
I am pleased to inform you that after reviewing the updated manuscript, I have no further specific comments to add. The revisions meet the expectations outlined in my previous comments. The adjustments you have made have significantly improved the clarity and overall impact of the study. I wish all of the authors success in their future research and look forward to their continued contributions to the field.

Sincerely,